# Elaboration and Characterization of a Biochar from Wastewater Sludge and Olive Mill Wastewater

Younes Gaga [1], Imane Mehdaoui [2], Mohammed Kara [1], Amine Assouguem [3,4,*], Abdulrahman Al-Hashimi [5], Mohamed Ragab AbdelGawwad [6], Mohamed S. Elshikh [5], El Mokhtar Saoudi Hassani [2], Mona S. Alwahibi [5], Jamila Bahhou [1], Mustapha Taleb [2] and Zakia Rais [2]

1 Laboratory of Biotechnology, Conservation and Valorisation of Natural Resources (LBCVNR), Faculty of Sciences Dhar El Mahraz, Sidi Mohamed Ben Abdallah University, Fez 30000, Morocco
2 Laboratory of Electrochemistry, Modelisation and Environnment Engineering, Faculty of Sciences Dhar El Mahraz, Sidi Mohamed Ben Abdallah University, Fez 30000, Morocco
3 Laboratory of Functional Ecology and Environment, Faculty of Sciences and Technology, Sidi Mohamed Ben Abdellah University, Fez 30000, Morocco
4 Laboratory of Applied Organic Chemistry, Faculty of Sciences and Technology, Sidi Mohamed Ben Abdellah University, Imouzzer Street, Fez 30000, Morocco
5 Department of Botany and Microbiology, College of Science, King Saud University, Riyadh 11451, Saudi Arabia
6 Genetics and Bioengineering, Faculty of Engineering and Natural Sciences, International University of Sarajevo, 71210 Sarajevo, Bosnia and Herzegovina
* Correspondence: assougam@gmail.com

**Abstract:** The objective of this study is to valorize two waste products which, until now, caused major problems concerning their management and impacts on the environment and health. This study concerns the sludge of the wastewater treatment station of the city of Fez-Morocco and the olive mill wastewater, which are produced, respectively, in quantities of around 51,100 t/year and 514,350 m$^3$/year, by pyrolysis for the production of biochar. The obtained biochar was characterized by physicochemical and spectroscopic analyses. The results show that the biochar is close to neutrality and is characterized by an important organic and mineral load; further, it is endowed with a porous surface, which could facilitate the adsorption of different polluting substances, composed mainly by micropores. It is mainly composed of alcohol, phenol, carboxyl and phenyl groups, as well as other mineral elements including silica and calcite. The composition, structure and morphology of the biochar thus prepared recommend its use in various fields, such as the treatment of pollutants, organic amendment, the reinforcement of polymers and as a secondary building material.

**Keywords:** sewage sludge; lagoons; biochar; pyrolysis; characterization; physicochemical; surface; spectroscopic

## 1. Introduction

Population growth, industrialization, agricultural development and unthinking modernization are among the main sources of environmental pollution [1,2] in the form of the excessive amounts of solid and liquid waste generated and not treated or recovered [3], which can directly impact human health and the environment.

In addition, according to the United Nations Organization (UNO), by 2030, the need for water will increase by almost 50% [4], which will lead to the reduction of water resources and make it more difficult to access [5]. However, this crisis can also influence other sectors; in particular, the needs of the agricultural sector.

On the one hand, the quantities of wastewater produced from different sources—domestic, industrial, commercial or agricultural [6]—further aggravate the environmental situation.

In this regard, the National Office of Electricity and Drinking Water (NOEDW) aims to increase the number of wastewater treatment stations (WWTS) to 164 by 2023 for a total capacity of nearly 530,000 m$^3$/d [7], to treat and minimize the impact of wastewater on the environment. This is accompanied by the production of huge quantities of foul-smelling sludge. The forecasts are for a national production of 300,000 tons/year by 2025 and 500,000 tons in 2030 (ONEP, Morocco) [8], with a quantity of approximately 150 to 200 kg/year per inhabitant equivalent [9]. Hence, there is need for well thought out management of these wastes due to their increasing volume and the risks of pollution they generate [8].

In Morocco, no national program (PNA, DMA, DD...) encourages the definition of a regulation on the management of sludge from wastewater treatment stations [10]. Currently, they are either dumped or recovered (73%) by spreading on agricultural land as an organic amendment despite their high pollution load and dryness [11,12].

On the other hand, Morocco is among the Mediterranean countries producing olive oil; it currently has a cultivated area of 998,000 hectares and an annual yield of 1,143,000 tons of olives generating 514,350 m$^3$ of olive mill wastewater [13]. According to the latest statistics of the Regional Directorate of Agriculture of Fez-Meknes, the region Fez-Meknes monopolizes more than 36% of the national production of olives accompanied by 185,166 m$^3$ of olive mill wastewater [14], of which less than 30% is transformed into solid waste by natural evaporation. The rest is discharged into rivers, causing serious environmental and technical problems that arise from the malfunction of wastewater treatment stations during the period of olive crushing (November-February); this is due to the production's high acidity and large loads of settleable, non-biodegradable organic matter, including polyphenols [14].

There are several studies on the treatment and valorization of these two types of waste, in particular on the production of biochar which is a solid material resulting from a pyrolysis, a thermal treatment [15] based on the action of heat in an inert atmosphere (no oxidation or addition of other reagents) which allows us to obtain a carbonaceous solid, i.e., the biochar, an oil and a gas, i.e., the wood distillate [15,16]. Pyriolysis starts at a relatively low temperature level (200 °C) and continues up to about 1000 °C or more.

Indeed, the biochar could be made from different sources, e.g., animal matter such as manure chicken [17,18], bovine bones [19], fish waste [20] such as shellfish [21] and fish scales [22], or vegetal waste [23,24], wood waste [25,26] or biomass [27].

In addition, the biochar can be used in many fields, especially in agriculture; in fact, it can be used to promote soil fertility and increase agricultural yields [28] as well as reduce salt stress [29] and detoxify the soils for the cultivation of lettuce [16,30].

The objective of this investigation is the elaboration of a biomaterial "biochar" based on olive mill wastewater and the sludge of wastewater treatment stations, and the determination of its physicochemical and spectroscopic characteristics that determine the areas of its further use outside the realm of agriculture.

## 2. Materials and Methods

### 2.1. Materials

To realize this work, we used sludge supplied by the wastewater treatment station of the city of Fez (STEP) which was stabilized by anaerobic digestion, dehydrated by filter band and brought to 80% of dryness by solar drying. The olive mill wastewater was issued from a traditional olive crushing industry in the industrial quarter DOKKARAT of the city of Fez.

Both wastes are conditioned, transported, stored and analyzed according to the standards approved by AFNOR NF EN 14742 [31] for sludge and enacted by Rodier for wastewater [32].

### 2.2. Preparation of Biochar

In a 100 L forced action mixer, Soroto model, a mass of sludge of 80% dryness was mixed for 15 min with a volume of OM wastewater in order to reproduce a dryness of 50% mixture of sludge and OM wastewater and to enrich the mixture in carbon owing to the richness of the OM wastewater in organic and mineral matters. The obtained mixture was dried again naturally or in a forced way in order to increase again its dryness to 80%. The dry powder was slowly pyrolyzed, in the absence of oxygen, at 500 °C for 4 h in a tubular furnace opening to 1700 °C, model HTRV-A 17/70/250. The cooled product constitutes the biochar. This last one was analyzed by analytical and spectroscopic methods to foresee its use in the depollution of the environments.

It should be noted that in the case where the olive mill wastewater was supplied from a modern crushing unit where there is the addition of water in the crushing process, it is necessary to take into account the dilution factor in order to obtain the expected results or to reach the desired concentrations in carbon rate and/or mineral matter in the finished product, i.e., the biochar.

### 2.3. Physicochemical Characterization of Biochar

The physicochemical characterization of the biochar was carried out by determining the pH (ISO 10390-2005) with a pH meter type HANNA pH 209, the electrical conductivity (ISO 11256-1994) with a conductivity meter type HANNA EC 214, the humidity of the dry matter (ISO 11465-1993) with an oven type BOXUN, the organic matter (NF EN 13039-2011) with a muffle furnace type Barustead Thermolyne 1400 °C. Total organic carbon and mineral matter was deduced from the value of organic matter by a calculation according to AFNOR standards. Total nitrogen Kjeldhal (ISO 11261-1995) was determined with a SELECTA nitro-pro type distiller, fat content (NF EN ISO 734-1, 2000), and polyphenol content was determined according to the Folin–Ciocalteu method (ISO 14502-1-March 2005) with a spectrophotometer model UV-1800PC UV/VIS.

Trace elements and metallic trace elements are determined by inductively coupled plasma atomic emission spectrometry (ICP-AES) model Activa of Horiba Jobin-Yvon, equipped with an argon plasma.

It should be noted that the fat (MG), polyphenols (Phy) and metallic trace elements (ETM) were analyzed to verify the non-toxicity of the biochar because the raw materials which constitute it are waste.

### 2.4. Surface Chemical Characterization of Biochar

This was accomplished by evaluating the point of zero charge (PZC), performed by the method developed by Kalay et al. [33]. The porous texture was studied by the adsorption of methylene blue (BM) according to Pelekani and Snoeyink [34]. The measurement of the indices of methylene blue was performed by spectrophotometry reported by Hameed et al. [35] Iodine was determined by ASTM D4607- 94 [36], deduced from the AWWA standard defined by Robinson and Hansen [37]. The type of surface functions were analyzed by Boehm's method [38], extracted from the work of Baudu et al. [39].

### 2.5. Spectroscopic Characterization of Biochar

Spectroscopic characterization was accomplished by analysis of the biochar by Fourier transform infrared spectroscopy (FTIR), X-ray diffraction (XRD) and scanning electron microscopy (SEM) coupled with an EDX probe and UV spectroscopy.

FTIR analysis was performed using a Bruker (Germany) Vertex 70 IR-TF spectrophotometer, in the range of 400–4000 cm$^{-1}$, using the ATR mode, and accumulating 16 scans with a resolution of 4 cm$^{-1}$.

The DRX diffractogram of the PM powder was obtained by PAN-Critical X' Pert Pro X-ray diffractometer equipped with a monochromatic Cu-K$\alpha$ source (1.54 Å), operating at a voltage of 40 kV and a filament current of 30 mA to assess the crystallinity of the biochar. The DRX was recorded with a continuous scan from 5° to 80° with a scan step time of

34 s. Abundant chemical components in the material are predicted by processing via the International Diffraction Data Center (IDDC) powder diffraction database.

Scanning electron microscopy (SEM) was used to determine the surface morphology of the biochar, and via an environmental scanning electron microscope equipped with an EDX probe, model QUANTA 200 coupled to energy dispersive spectroscopy (EDS) at an accelerating voltage of 15 kV.

Thermogravimetry (TGA) was used to determine the absolute mass loss of biochar as a function of temperature. It was operated using a LINSEIS STA PT1600 instrument. A mass sample of 18.9 mg was introduced into an alumina crucible supported on a balance located in the instrument's oven. The analysis was performed in an air atmosphere with a ramp of 10 °C/min in the temperature range of 30 °C to 1000 °C.

## 3. Results and Discussions

### 3.1. Physicochemical Characterization of the Biochar

Physicochemical analyses (Table 1) show that biochar is almost neutral and non-toxic [40] due to its negligible load of polyphenols, fatty matter (FM) and metallic trace elements (Table 2). It has a high dryness (80%), an important mineral composition favoring ionic exchange and heat transfer and thus can be used as a thermal conductor [41] or as an addition to the manufacture of construction materials such as bricks and paving stones.

**Table 1.** Physicochemical characteristics of biochar.

| Parameters | pH | EC (μS/cm) | H (%) | DM (%) | MM (%) | OM (%) | Polyphénols (%) | FM (%) | TOC (%) | NTK (%) | C/N |
|---|---|---|---|---|---|---|---|---|---|---|---|
| Values | 6.65 | 1300 | 1.58 | 98.42 | 78.31 | 21.69 | 1.09 | 2.16 | 12.58 | 1.05 | 11.98 |

**Table 2.** Trace element and metallic trace element contents of biochar.

| Elements | Al | Ca | Cu | Fe | K | Mg | P | Mn | Na | Zn |
|---|---|---|---|---|---|---|---|---|---|---|
| Concentration (mg/g) | 4.9396 | 64.009 | 0.2535 | 7.4737 | 3.6947 | 18.646 | 22.588 | 0.3118 | 25.336 | 0.7603 |
| **Elements** | **B** | **Ag** | **Cd** | **Co** | **Cr** | **Ni** | **Pb** | **Se** | **Ti** | **As** |
| Concentration (mg/g) | <0.01 | 0.0458 | <0.01 | <0.01 | 1.0165 | <0.01 | <0.01 | <0.01 | <0.01 | <0.01 |

In addition to its richness in trace elements (Table 2), including calcium (64.01 mg/g), sodium (25.34 mg/g), potassium (22.59 mg/g), magnesium (18.65 mg/g), iron (7.47 mg/g), aluminum (4.44 mg/g), potassium (3.70 mg/g) and the presence, in decreasing order, of chromium, zinc, copper and silver, the biochar has an average organic composition (Table 1) of which more than half is in the form of organic carbon and almost 1% in total nitrogen Kjeldhal, leading to a C/N ratio between 10 and 15. This margin is recommended for its use as an organic soil amendment in difficult climates, i.e., those poor in mineral and organic elements.

The result obtained corroborates several research works that have shown the fertilizing efficiency of biochar in combination with other organic amendments [42–44].

The acquired results could indicate the use of this biochar as an organic amendment because of its richness in trace elements (Ca, Na, P, Mg, K, Fe, Zn, Mn, Cu, Zn and Ag) for soils poor in these elements.

### 3.2. Surface Characterization of Biochar

#### 3.2.1. Zero Charge Point pHpzc of Biochar

The pH of zero charge (pHpzc) corresponds to the pH of the solution for which the curve C, representing the final pH versus the initial pH of the solution (Figure 1), crosses the first bisector B (final pH = initial pH).

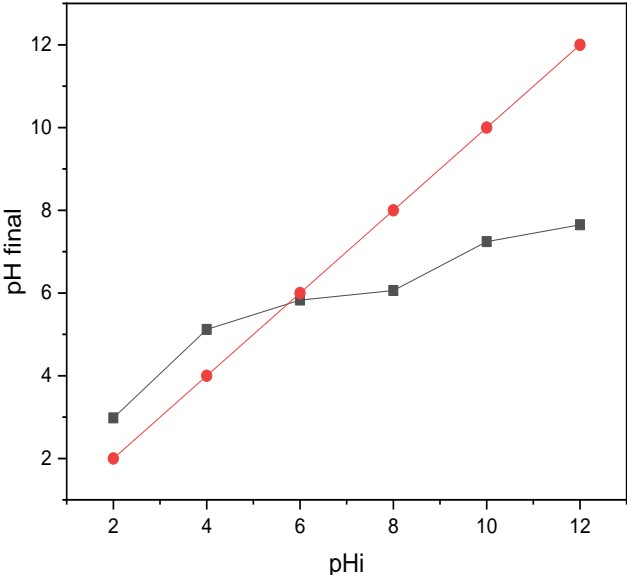

**Figure 1.** Determination of the pH of zero charge (pHpzc) of the biochar.

It is relative to the pH of the aqueous solution in which the biochar exists under a neutral electric potential. It is equal to 5.80, thus explaining why the biochar surface is positively charged for pH of the medium lower than this value; it is thus protonated and acidic. In addition, for a pH of a medium higher than the pHpzc, the biochar surface is negatively charged and becomes deprotonated and basic. This type of surface could be favorable for the adsorption of anionic pollutants (pH < pHpzc), as well as cationic (pH > pHpzc).

3.2.2. Specific Surface of Biochar by the Methylene Blue Method (BM)

The mesoporous volume of biochar was estimated by adsorption experiments of the cationic dye BM. The adsorption isotherm was established by stirring 0.5 g of biochar for 6 h in 50 mL of BM solution at initial concentrations ranging from 20 to 100 mg/L. The mixture was centrifuged at 6000 rpm. The absorbance of the filtrate leads by extrapolation to the residual concentration of the BM at equilibrium ($Q_e$) via the following equation:

$$Q_e = (C_o - C_e) \times V/m \tag{1}$$

with $Q_e$ (mg/g) and $C_e$ (mg/L) denoting, respectively, the amount of substance adsorbed per gram of biochar and the concentration of BM in solution at equilibrium.

The adsorption isotherm of BM was simulated by the linear equation of the Langmuir model [45]:

$$\frac{C_e}{Q_e} = \frac{1}{K_L Q_{max}} + \frac{C_e}{Q_{max}} \tag{2}$$

with $C_e$ (mg·L$^{-1}$): residual solution concentration at adsorption equilibrium;

$Q_e$ (mg·g$^{-1}$): amount of adsorbate adsorbed at equilibrium;

$K_L$ (L·mg$^{-1}$): adsorption equilibrium constant, of the BM/biochar couple, of the Langmuir model;

$Q_{max}$ (mg·g$^{-1}$): maximum amount of BM adsorbed.

The wavelength of the dye absorption maximum (Figure 2) is 663 nm.

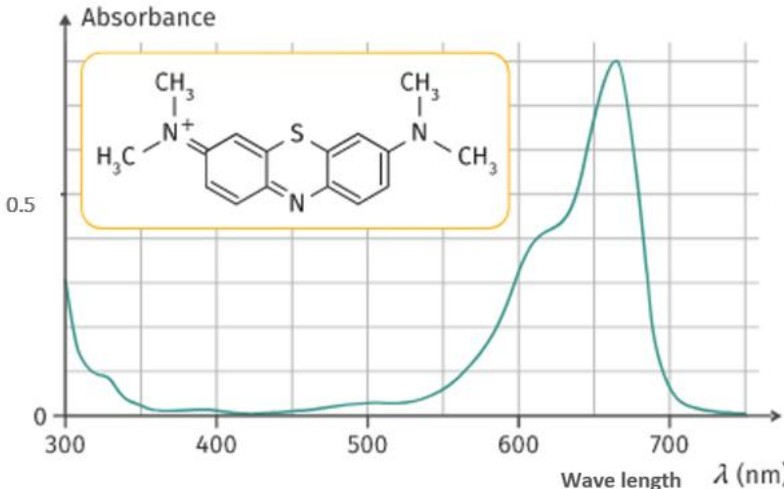

**Figure 2.** Absorption spectrum of methylene blue in the visible ($C_0 = 10$ mg·L$^{-1}$).

The calibration curve for methylene blue for concentrations between 0 and 10 mg/L is as follows (Figure 3), the equilibrium isotherm and Langmuir isotherm are presented in Figures 4 and 5, respectively.

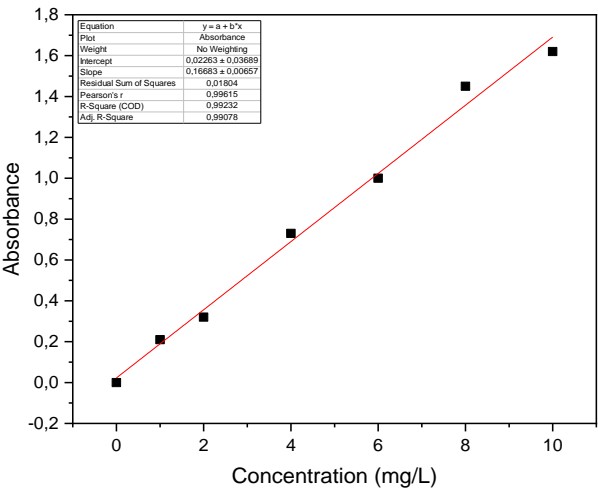

**Figure 3.** BM calibration line; at $\lambda_{max} = 669$ nm.

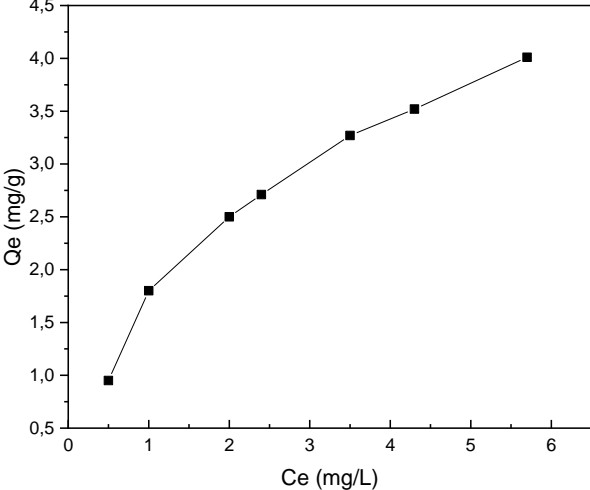

**Figure 4.** Equilibrium isotherm.

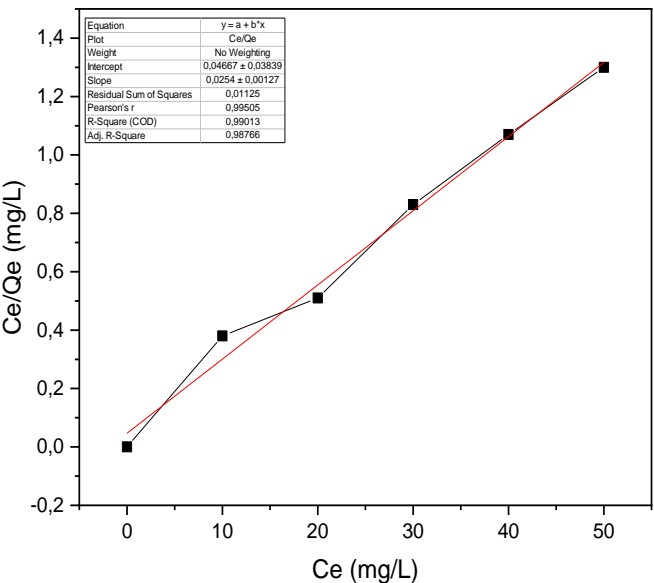

**Figure 5.** Langmuir isotherm.

The specific surface area of the biochar covered by the BM molecule (SBM) is evaluated from the $Q_{max}$ of the adsorbed BM following the equation:

$$S_{BM} = Q_{max}A_{BM}\left(\frac{6.02 \times 10^{23}}{M_{BM}}\right) \tag{3}$$

with:

$A_{BM}$: molecular surface area of BM = 1.30 nm$^2$ [46];
$M_{BM}$: molar mass of BM = 319.85 g·mol$^{-1}$;
$Q_e$: Quantity adsorbed at equilibrium (mg·g$^{-1}$);
$C_e$: Concentration of the solution at equilibrium (mg·L$^{-1}$).

The calculated surface area of the studied biochar is 694.71 m$^2$/g. This encourages its exploitation as an adsorbent material for environmental pollutants.

### 3.2.3. Methylene Blue Index (MBI)

The IBM of the elaborated biochar is evaluated by measuring the absorbance at 620 nm via a spectrometer of a diluted filtrate obtained by the stirring of 1 g of biochar in 25 mL of methylene blue solution of concentration 1.20 g/L for 30 min (Hameed et al., 2007), and expressed in mg of adsorbed methylene blue per g of biochar according to the expression:

$$I_{BM} = \left[\left(\frac{C_i - C_r \times m_b}{m_b}\right)\right] \times V \tag{4}$$

with:

$C_i$: Initial concentration of the solution (1.2 g/L);
Cr: Residual concentration after 30 min of agitation (mg/L): Mb: Mass of biochar (1 g);
V: Volume of BM (25 mL).

The IBM of the biochar is 9.2 mg/g, proving that the mesoporous surface is capable of adsorbing some molecules of medium size (2–50 nm) and sizes larger than 50 nm.

### 3.2.4. Iodine Index ($I_i$)

$I_i$ is obtained by stirring 0.2 g (Mb) of biochar in 10 mL of 5% hydrochloric acid, boiled for 30 s, to which was added 20 mL ($V_{ads}$) of 0.1 N iodine ($C_0$) while maintaining stirring for 30 s. The mixture was then filtered. Of the filtrate, 10 mL ($V_{I_2}$) is titrated with 0.1 N sodium thiosulfate ($C_n$) in the presence of starch starch.

$I_i$ is the amount of iodine absorbed (mg) per g of biochar) according to the equation:

$$I_i \left( mg \cdot g^{-1} \right) = \frac{\left[ C_0 - \frac{C_n V_n}{2 V_{I_2}} \right] M_{I_2} V_{ads}}{M_b} \tag{5}$$

with:

$V_n$: Volume of sodium thiosulfate at equivalence (mL);

$M_{I_2}$: Molar mass of iodine (254 g·mol$^{-1}$).

The $I_i$ of the studied biochar reveals that it is microporous (462.35 mg·g$^{-1}$). This explains why the internal surface of the micropores of this material could retain small molecules (<2 nm) [46] of polluting water, atmosphere or soil.

### 3.2.5. Surface Chemical Characteristics of Biochar

Oxygen and basic group contents are determined by mixing 0.15 g of the biochar with 50 mL of a 0.1 mol·L$^{-1}$ aqueous solution of each reagent (NaOH, Na$_2$CO$_3$, NaHCO$_3$, NaOC$_2$H$_5$, HCl). The solutions are stirred for 48 h and then filtered. Of each filtrate, 30 mL is determined by pH-metry. The basic solutions are assayed by HCl (0.1 mol·L$^{-1}$), the acidic solution by NaOH (0.1 mol·L$^{-1}$).

The results show that the content of oxygenated functional groups in the biochar is 9.6%. The corresponding Boehm assays (Table 3) exhibit fewer basic groups (lactones and carbonyls) than acidic groups (carboxyl's and phenolics), indicating its slightly acidic character.

**Table 3.** Surface chemical characteristics of the elaborated biochar.

| Groups | Carboxylic | | Phenolics | Lactones | Carbonyls | Total Oxygenates | Total Bases |
|---|---|---|---|---|---|---|---|
| Value (meq/g) | 0.007 | | 1.814 | 0.005 | 2.302 | 0.018 | 0.416 |
| **Parameters** | **pH$_{PZC}$** | **S$_{BM}$ (m$^2$/g)** | **Oxygenated Functional Groups (%)** | **Iodine Index (mg/g)** | | | **BM Index (mg/g)** |
| Value | 5.56 | 694.71 | 9.6 | 462.35 | | | 9.22 |

### 3.3. Spectroscopic Characterization

#### 3.3.1. Fourier Transform Infrared

The analysis by Infrared spectroscopy was carried out in order to determine the functional groups, and the bonds developed during the preparation of the biochar. The FTIR spectrum (Figure S1) was scanned between 4000 and 500 cm$^{-1}$ and confirms the existence of several chemical functions: O-H C=O, C=C, Si-O et CaCO$_3$.

The infrared spectrum results of the biochar show a bond at 3300 cm$^{-1}$ of low intensity corresponding to the O-H stretching vibration, probably attributed to the polyphenols commonly found in olive mill wastewater [47,48]. The band observed near 1750 cm$^{-1}$ is probably related to C=O stretching, due to the carboxylic and carboxylate form. Moreover, the peak around 1450 cm$^{-1}$ could be attributed to the C=C vibrational mode of the phenyl groups; this is qualitatively confirmed with those deduced from the study of surface functions revealing the notable presence of lactone and phenol functions. While the pronounced peaks around 1000 cm$^{-1}$ can be attributed to the elongation vibrations of C-O bonds characteristic of alcohols, this peak can also be attributed to the elongation vibrations of Si-O bonds in silica [49]. The peak extending to 860 cm$^{-1}$ is due to the presence of calcite (CaCO3) [50].

#### 3.3.2. X-ray Diffraction

The diffractogram of biochar presented in Figure S2 relates that the main diffraction peaks are located in the 2θ zone from 20° to 70°. Indexing of the most intense peaks revealed the presence of three crystalline phases corresponding to calcium carbonate (CaCO$_3$), silicon

dioxide ($SiO_2$) and dolomite ($CaMg (CO_3)_2$), respectively, identified via comparison with database files reference code ICDD 01-086-0174, ICDD 01-078-1252 and COD 96-900-3514.

The peaks of $CaCO_3$, in Rhombohedral form, are located at $2\theta$ = 20.89, 23.14, 26.68, 29.47, 30.99, 36.06, 39.52, 41.22, 43.26, 47.58, 48.59 and 50.20°. Those of $SiO_2$, in the hexagonal crystal form, are located at $2\theta$ = 20.89, 23.14, 26.68, 29.47, 30.99, 36.06, 39.52, 41.22, 43.26, 47.58, 48.59 and 50.20°. On the other hand, the peaks of $CaMg (CO_3)_2$, crystallized in hexagonal form, are located at 20.89, 23.14, 26.68, 29.47, 30.99, 36.06, 39.52, 43.26, 47.58, 48.59 and 50.20. The result obtained is in agreement with a study of phosphorus adsorption on a biochar substrate moistened with olive mill wastewater [51].

### 3.3.3. Scanning Electron Microscopy Coupled to an EDX Probe

SEM images at different magnifications (Figure 6) show that the surface of the BY1B biochar is composed of condensed aggregates of small, spaced particles. Moreover, the visible, rough and irregular surface of the material reveals its porous nature.

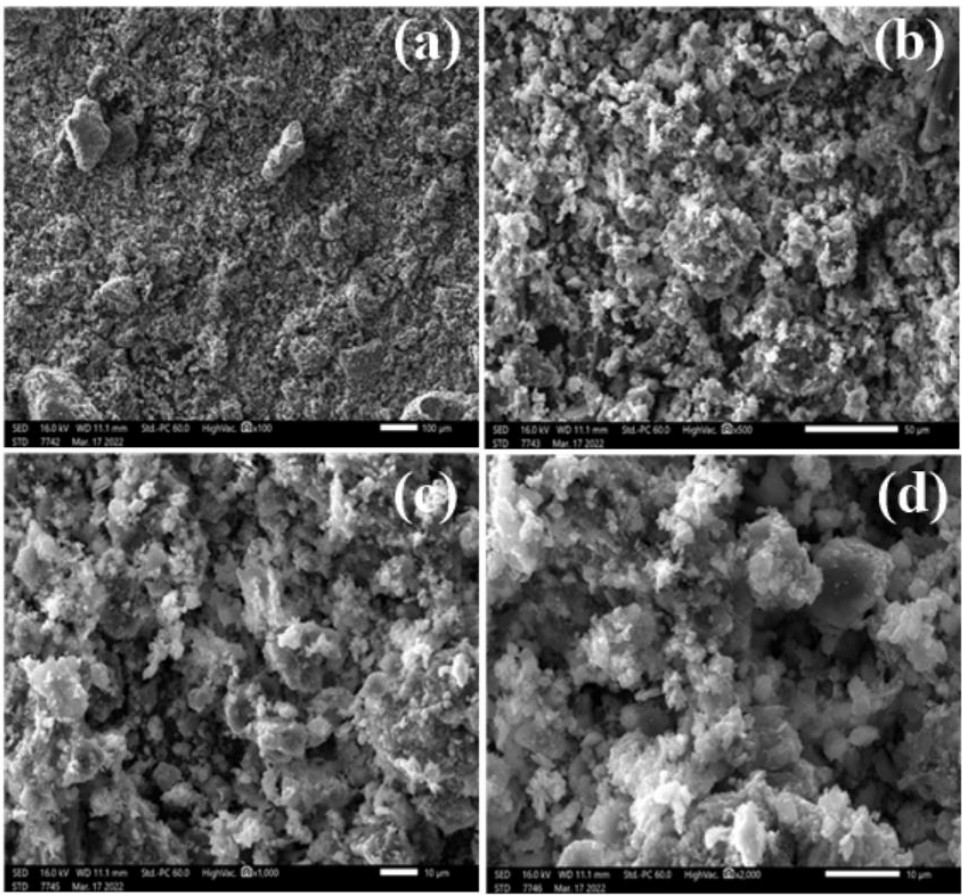

**Figure 6.** SEM images of biochar at different magnifications (**a**) ×100, (**b**) ×500, (**c**) ×1000 and (**d**) ×2000.

This character is considered as a factor that favors the adsorption of pollutants from the environment by ion exchange mechanisms, and that leads to the fixation of cationic or anionic particles. Indeed, several studies have used biochar as an adsorbent to remove medical substances such as levofloxacin from aquatic environments as well as the reduction of heavy metals such as arsenic, phosphate, lead and copper and the decolorization of wastewater from textile industries. Furthermore, the biochar has the potential to have a beneficial effect on soil contaminated by hydrocarbons [30,52,53].

Elemental chemical microanalysis of the biochar surface allows for the quantification of its major composition in mass and atomic content [54]. This is presented in the EDX spectrum (Figure 7) and Table 4.

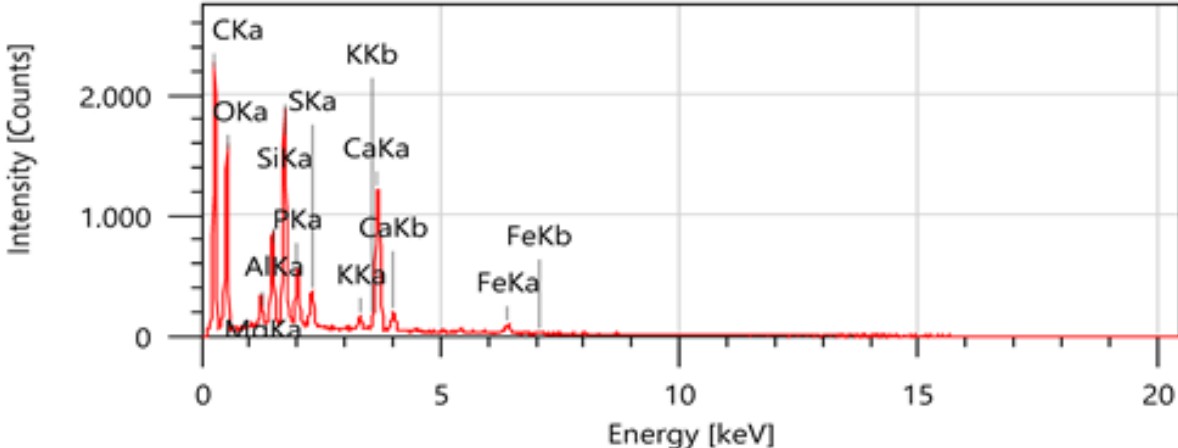

**Figure 7.** EDX spectrum of biochar.

**Table 4.** Mass and atomic percentages of the different elements of the biochar.

| Elements | Mass Percentage | Atomic Percentage |
|---|---|---|
| C | 43.59 ± 0.15 | 57.90 ± 0.21 |
| O | 29.24 ± 0.25 | 29.15 ± 0.25 |
| Mg | 0.97 ± 0.03 | 0.64 ± 0.02 |
| Al | 2.78 ± 0.05 | 1.64 ± 0.03 |
| Si | 6.60 ± 0.07 | 3.75 ± 0.04 |
| P | 2.53 ± 0.04 | 1.30 ± 0.02 |
| S | 1.47 ± 0.03 | 0.73 ± 0.02 |
| K | 0.63 ± 0.03 | 0.26 ± 0.01 |
| Ca | 10.26 ± 0.10 | 4.08 ± 0.04 |
| Fe | 1.94 ± 0.07 | 0.55 ± 0.02 |
| Total | 100.00 | 100.00 |

The spectrum relates the presence of intense peaks related to the elements carbon, oxygen, silicon, phosphorus, sulfur, calcium, aluminum, iron and potassium.

The major mass percentages of biochar are noted for carbon, oxygen, calcium and silicon at 43.59% and 29.24%, 10.26%, 6.60%, respectively. The other elements, adding up to 10.31%, constituted, in descending order, aluminum, phosphorus, iron, sulfur, potassium and magnesium.

The same ranking was recorded for the atomic percentage; the elements strongly present are carbon with 57.90, oxygen with 29.15, calcium with 4.08 and silicon with 3.75. The other elements—aluminum, phosphorus, sulfur, magnesium, iron—constitute the remaining 5.12%.

The results obtained seem to be due to the presence of different oxides, such as $Al_2O_3$, FeO, MgO, CaO, $SiO_2$, $Na_2O$, $K_2O$, etc.

### 3.3.4. Thermal Analysis

Thermal analysis of the biochar, shown in Figure 8, shows the variation in mass over a temperature range of 0 to 1000 °C.

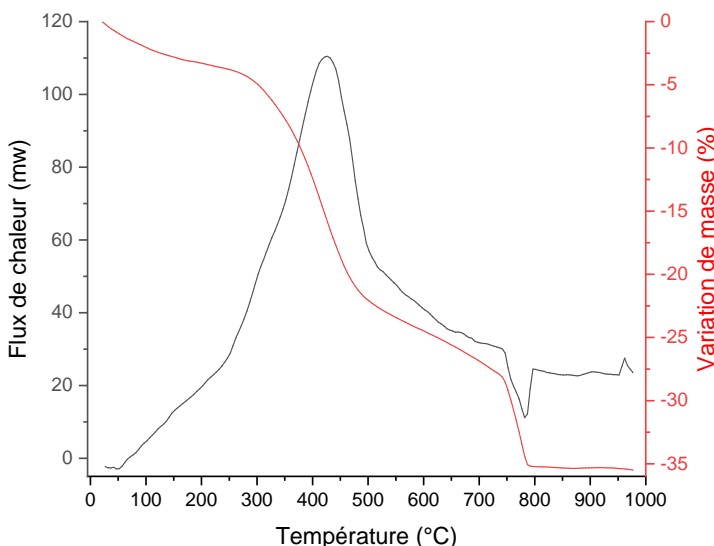

**Figure 8.** Thermogravimetric analysis (ATG) spectrum of biochar.

The global mass loss was unfolded in four steps: The first loss is located between 0 and 300 °C; we noted a slight loss of 4% that could correspond to the traces of trapped water molecules (seeing that the biochar was obtained by pyrolysis). The second loss begins at 300 °C to end at 500 °C accompanied by an exothermic peak, while the third is in the vicinity of 750 °C associated with an endothermic peak. These can be attributed to the degradation of $CaCO_3$ into calcium oxide ($CaO$) and carbon dioxide ($CO_2$), according to the following reaction [55]:

$$CaCO_{3(solid)} \leftrightarrow CaO_{(solid)} + CO_{2(gaz)}. \tag{6}$$

Calcite is considered to be a major component in the studied biochar. Above 800 °C, no mass loss was noticed. Indeed, an amount of 64% remains after the treatment at 1000 °C, which shows that the biochar has an organic character.

### 4. Conclusions

In this study, our research into a new material, biochar—originating from two precursors, sewage sludge and olive mill wastewater—allowed us to predict some potential fields of use. We could mention its application as an organic amendment to promote the fertility of soils lacking in mineral elements since its organic matter and trace element content is very high, and as a thermal conductor thanks to its high electrical conductivity. In addition, its porous morphology supports its use as an adsorbent of several pollutants exhibited in wastewater metals, dyes or in gases produced by combustion and biomethanization, including $H_2S$, $SO_x$ and $NO_x$. Furthermore, the material's high dryness suggests its possible application as a by-product in the manufacture of construction materials.

**Supplementary Materials:** The following are available online at https://www.mdpi.com/article/10.3390/su15032409/s1, Figure S1: Infrared spectrum of Biochar; Figure S2. X-ray spectrum of Biochar (BY1B).

**Author Contributions:** Y.G., A.A.-H. and A.A. were charged with the writing and submission; I.M., M.R.A., E.M.S.H. and M.S.A. were charged with the funding and interpretation; M.K., A.A., M.S.E. and J.B. were charged with the supervision; M.T. and Z.R. were charged with the direction. All authors have read and agreed to the published version of the manuscript.

**Funding:** This work was funded by the Researchers Supporting Project (number RSP2023R173), King Saud University, Riyadh, Saudi Arabia.

**Data Availability Statement:** Data will be available on request.

**Acknowledgments:** The authors extend their appreciation to the Researchers Supporting Project (number RSP2023R173), King Saud University, Riyadh, Saudi Arabia. The research work that gave rise to this publication was carried out thanks to the contribution and synergy of all the authors. We would like to thank the Director General of RADEEF, Fatima Guennouni, who allowed us to visit the waste sites and to supply important quan-tities of samples through the STEP and depollution division of the city of FEZ.

**Conflicts of Interest:** The authors declare that they have no know competing financial interests or personal relationships that could have appeared to influence the work reported in this paper.

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
