# Peer review of "Elaboration and Characterization of a Biochar from Wastewater Sludge and Olive Mill Wastewater"

_sustainability, doi:10.3390/su15032409_

Round 1
Reviewer 1 Report
The submitted manuscript fits well to the journal’s profile. The presented research described a biochar, a new product obtained in a process of pyrolysis from wastewater sludge and olives mill wastewater. The obtained results are an important offer to the international environmental problem solution of the waste storage and utilization.
The paper is structured very well and its quality is high also easy to understand by the readers.
Nevertheless I suggest few changes.
Introduction
Line 68 “In Morocco, there is no national program (PNA, DMA, DD...) *” Could you explain this sentence?
Line 84, it would be a huge advantage if in this line you add a paragraph (about 3-4 sentences) concerning a process of pyrolysis. You can base on the literature:
Coloured sintered glass-ceramics from hospital incineration fly ash, Materials Letters, 2019, vol. 252, 34-37
Vitrification of medical waste as an alternative method of wastes stabilization, 19 (12a), Fresenius Environmental Bulletin, 2010, 3045-3048
Materials and methods.
Line 100 “15 mn” please change to “15 min” or “15 minutes”.
Results and discussions
Table 1, table 2, table 3 and table 4. Please, change all the commas in the results numbers to points for example 6,65 to 6.65.
Please, unify the scale of the figures 3, 4 and 5 as well as pictures on figure 8.
Would be possible to change the language of figure 10?
Conclusions
Line 350. This part summarizes your results so please do not write references in this section. Why did you write “…” in line 350 and line 351?
I suggest to write few more time the manuscript to avoid the mistakes.
Author Response
The authors would like to express their gratitude to the reviewers for the revision of our manuscript entitled: “ Synergistic effect of bioactive monoterpenes against the mosquito, Culex pipiens (Diptera: Culicidae)”. We sincerely appreciate all valuable comments and suggestions.
Our responses to the Reviewers’ comments are described below. Appropriated changes, suggested by the Reviewers have been introduced to the manuscript (highlighted within the document).
Introduction |
- Line 68” In Morocco, there is no national program (PNA, DMA, DD) *” Could you explain this sentence? No Moroccan program addresses the regulation of sludge from wastewater treatment plants (the sentence has been reformulated in the text) - Line 84, it would be a huge advantage if in this line you add a paragraph (about 3-4 sentences) concerning a process of pyrolysis
|
Materials and Methods |
- Line 100 “15 mn” please change to “15 min” or “15 minutes”. “15 mn” has been changed by “15 min (Line 102)
|
Results and discussions |
- Table 1, table 2, table 3 and table 4. Please, change all the commas in the results numbers to points for example 6,65 to 6.65. All commas have been replaced by periods in all values of the manuscript. - Please, unify the scale of the figures 3, 4 and 5 as well as pictures on figure 8. The scale had been modified - Would be possible to change the language of figure 10? ‘Thermogravimetric analysis’ has been added
|
Conclusion |
- Line 350. This part summarizes your results so please do not write references in this section. Why did you write” …” in line 350 and line 351? The references have been removed |
Sincerely
Reviewer 2 Report
The MS needs to be improved in general. The introduction needs to be implemented because this is not enough. All the Figures need to be changed and made more presentable, in this way they are not adequate for scientific work, clarity needs to be made even on the figures not only on the written part of the MS. Several comments are in the text, please authors to follow them to improve the work in general. All references should be written following the guidelines of the journal Sustanability.

Author Response
Dear Reviewer,
The authors would like to express their gratitude to the reviewers for the revision of our manuscript entitled: “Elaboration and Characterization of a Biochar from Wastewater Sludge and Olives Mill Wastewater”. We sincerely appreciate all valuable comments and suggestions.
Our responses to the Reviewers’ comments are described below. Appropriated changes, suggested by the Reviewers have been introduced to the manuscript (highlighted within the document).
Page 1 |
- Line 20: The email for corresponding author has been verified - Line 22: The sentence has been reformulated - Line 25: The “a” has been deleted
|
Page 2 |
- Line 47: The introduction has been expanded by adding 3 paragraphs (to line 74 from 87) and 17 references have been added - Line 48: All references have been numbered - Line 50: The sentence "This can cause direct impacts on human health and the environment" has been linked to the previous sentence in the text. - Line 52: Full name of UNO has been added in the text - Line 70 and 71: The reference has been added |
Page 2-3 |
- The material part has been rewritten (to line 94 from 99) |
Page 3 |
- Line 100: Olive mill has been modified by “OM” - Line 106: The “B” in majuscule had been modified by miniscule letter ’b’ - Line 132: All references have been modified according to the guideline of the journal |
Page 4 |
- Line 168: The points have been removed - Line 169: what value is this? mean? median? where is the error? The values mentioned in the table are mean values and are based on the results of three experiments. - Line181: The same answer as above (table 1) |
Page 6 |
- Line 218: Figure 3: The figure has been redone by another software - Line 220: Figure 4: The figure has been redone by another software The explanation of Ce and Qe were made in page 8 (to line 2140 from line 241) |
Page 7 |
- Line 222: Figure 5: The same answer as above (figure 4) The figure has been redone by another software |
Page 8 |
- Line 269: Figure 6: The figure has been inserted in a word document 'Supplimary material' S1 |
Page 9 |
- Line 289: Figure 7: The figure has been inserted in a word document 'Supplimary material' S2 |
Page 10 |
- Line 303: Figure 8 has been redone - The sentence and reference have been added (To line 328 from line 330 in the new version) |
Page 12 |
- Line 350: The points have been deleted |
Sincerely
Amine Assouguem
Round 2
Reviewer 2 Report
Some minor corrections to the text after which the article, in my opinion can be published. Authors look at the pdf file for the corrections.

Author Response
Dear Editor,
bellow the requested corrections,
Page 2 |
- Line 77: The sentence “which allows to obtain a carbonaceous solid, an oil and a gas” has been replaced by “which allows to obtain a carbonaceous solid, the biochar, an oil and a gas, the wood distillate”. And the reference was written as you suggested. Line 79: The number of all references has been modified - Line 83: The “…” had been modified by “.” - Line 86: “Akhtar et al., 2015” has been removed - Line 87: |
Page 11 |
- The word "gasoline" has been deleted (line 330)
|
Sincerely